# Thiol-Ene Photopolymerization and 3D Printing of Non-Modified Castor Oil Containing Bio-Based Cellulosic Fillers

**DOI:** 10.3390/polym17050587

**Published:** 2025-02-23

**Authors:** Rafael Turra Alarcon, Matteo Bergoglio, Éder Tadeu Gomes Cavalheiro, Marco Sangermano

**Affiliations:** 1Instituto de Química de São Carlos, Universidade de São Paulo-USP, São Carlos 13566-590, SP, Brazil; rafael.alarcon@usp.br (R.T.A.); cavalheiro@iqsc.usp.br (É.T.G.C.); 2Dipartimento Scienza Applicata e Tecnologia, Politecnico di Torino, Corso Duca degli Abruzzi 24, 10129 Torino, Italy; matteo.bergoglio@polito.it

**Keywords:** bio-based polymers, photopolymerization, biomass, 3D printing, renewable fillers

## Abstract

The photopolymerization process in 3D printing is considered greener once it involves a fast reaction and low energy consumption. Various reactions for photopolymerization can be used nowadays, but a special one is the thiol-ene “click” reaction that occurs in equimolar concentrations of thiol and alkene groups. In this sense, solvent-free photopolymerizable formulations were prepared to contain non-modified castor oil, Trimethylolpropane tris(3-mercapto propionate), and cellulosic fillers from hemp, tagua, and walnut. All formulations presented conversions higher than 70% and fast polymerization rates. Moreover, the filled formulations presented excellent curing depths in fewer seconds of light exposition, an important factor for their applicability in 3D printing. Furthermore, the hemp filler formulation presented the highest crosslinking density as determined by the DMTA, and was selected for printing two complex structures (pyramid and honeycomb shape). The rheology analysis showed that the formulations had adequate viscosities for the printer. Lastly, all polymers presented at least 97% bio-based contents, with gel contents superior to 96%.

## 1. Introduction

Polymers are unquestionably essential and ubiquitous in our society and can be used in various applications in numerous fields. However, concerns about large-scale production, non-renewability (i.e., using petroleum-based monomers), and discarding have been raised [1,2,3]. Therefore, developing fast, low-energy-consuming, solventless, and cleaner polymerization methods using renewable and low-toxic monomers is crucial. It can be called green polymerization design and is aligned with the Principles of Green Chemistry, Circular Economy, and United Nations Sustainable Development Goals—UNSDGs (e.g., Goal 9—Industry, Innovation, and Infrastructure; Goal 12—Responsible Consumption and Production; and Goal 13—Climate Action) [4,5,6,7,8,9].

In a greener methodology, the photopolymerization process can be a solution to avoid long reaction times and involve low energy consumption, since the reaction is driven by light [10,11,12,13,14,15,16,17]. This process has gained even more attention due to photocuring 3D printing (e.g., Stereolithography—SLA, digital light processing—DLP, and Continuous Liquid Interface Production—CLIP), where a chosen object can be produced layer-by-layer by curing the monomers in a lifting base [10,11,12]. After a layer formation, this base is elevated millimetrically to form a new layer; this process is repeated multiple times until the final production. The layer thickness and exposure time can be controlled, guaranteeing higher-quality architecture with no barbs or debris and high resolution [10,11,12].

Photopolymerization depends on the photoinitiators and correct monomers that are suitable to interact with the radicals/cationic species formed during the polymerization [10,11,12,13,14,15,16,17]. Vinyl, acrylated/methacrylated, epoxidized, and thiol monomers are some compounds that can be used in this process [10]. Thiol-ene is a useful reaction that can be applied in photopolymerization, in which a thiol compound (R-SH) reacts with a carbon–carbon double bond (Ene, R-C=C-R) in a step-growth process involving equimolar reaction and avoiding homopolymerization by the carbon-centered radical (i.e., no propagation step by the carbon-centered radical); this reaction is also considered as click chemistry, since it occurs in a fast way with high conversions [10,18,19,20,21,22]. This is true for alkenes and vinyl, but acrylate/methacrylate systems can still form homopolymers in the presence of thiol [23,24]. Therefore, alkenes are preferable for complete copolymerization with thiols, producing normally flexible polymers.

Compounds such as terpenes/terpenoids and vegetable oil derivatives have been used as renewable monomers for thiol-ene photopolymerization [25,26,27,28,29,30,31,32,33]. Vegetable oils are interesting compounds, since their structure contains a glycerol backbone with three fatty acid chains. Frequently, these chains contain carbon–carbon double bonds (unsaturated chains) [12,25]. Therefore, these groups can react directly with thiol or be modified to obtain epoxidized and further acrylated derivatives. However, the direct thiol-ene reaction does not easily occur in systems using vegetable oils, since some thiol compounds are not soluble/miscible and can take a long time to obtain the desired product [34,35,36]. Therefore, some authors prefer to use vegetable oil derivatives (i.e., epoxidized, acrylated, and maleinized) [30,37,38,39,40,41,42,43,44,45].

Castor oil was selected for this study to overcome the homogeneity issue, since it has mostly ricinoleic fatty chains containing hydroxyl groups (R-OHs) and alkenes [12,46]. The hydroxyl groups can assist the miscibility of some polar thiol compounds. Castor oil is a perfect choice considering its sustainability because it has a large production for technological proposals that do not compete with the food industry [12,46]. Despite its applicability, no study reported its use in the raw form in thiol-ene reactions; however, some used its derivatives. Therefore, castor oil can react with cysteamine, requiring a long reaction time, and the final product is reacted with epoxides. In other cases, the alkene can be reacted with 2-Mercaptoethanol to form polyalcohol compounds that are further reacted with isocyanates. Moreover, the OH presented in the ricinoleic chain can be reacted with thioglycolic acid or some other acid-containing OH, and these are further reacted with allyl bromide to provide alkenes, and then react with thiol [30,34,35,36,37,38,39]. As seen in all cases, castor oil should be previously modified before its use as a monomer.

Therefore, to our knowledge, this is the first attempt to use non-modified castor oil with a trithiol compound—Trimethylolpropane tris(3-mercaptopropionate)—by a photopolymerization process without solvent or dispersant agents. Using non-modified castor oil improves the greener and more sustainable aspect, since no modification steps are needed, therefore reducing the time and energy consumption and avoiding purification steps.

Cellulosic fillers have been used in acrylate and epoxidized vegetable oils due to their good dispersion and compatibility with these modified oils [47,48,49,50,51]. Therefore, bio-based cellulosic fillers from hemp, tagua, and walnut were added to the system to study their influence on the photopolymerization process and final properties of the crosslinked materials. These three fillers were selected due to their good interaction with epoxidized vegetable oils [52]. Finally, the best formulation was used for 3D printing to produce complex objects.

## 2. Materials and Methods

### 2.1. Materials

The castor oil (CO, leaking data: 03/2021; batch code: 12-60319) was acquired from Mundo dos Óleos (Brasília, Brazil). The trimethylolpropane tris(3-mercaptopropionate)—3SH (≥95.0%) and deuterated chloroform (CDCl_3_, 99.8% D) containing tetramethylsilane (TMS) were purchased from Sigma-Aldrich (São Paulo, Brazil) and used without further pre-treatment. The photoinitiator phenylbis(2,4,6-trimethylbenzoyl)phosphine oxide—BAPO (Irgacure 819; 97%)—was purchased from Sigma-Aldrich (Milano, Italy).

This study used three bio-based fillers, all kindly donated by Composition Materials Co., Inc. (Milford, MA, USA). The fillers were powders with 200 mesh from hemp (H), tagua (T), and walnut shell (W).

### 2.2. Castor Oil Characterization by ^1^H-NMR

First, the CO was solubilized in the CDCl_3_ and analyzed using an Agilent 400 MHz Premium Shiel (Bruker, Billerica, USA) spectrometer to obtain the ^1^H-NMR spectrum (Appendix A). The iodine value was calculated using Equation (1) by integrating the multiplet between 5.30 ppm and 5.44 ppm and the double triplet at 5.55 ppm (*K* is the sum of both integrations). However, to correctly integrate, the area should be normalized by the triple doublet at 2.31 ppm, which is related to 6 hydrogens of α-carbonyl in the fatty chain [53].(1)$$IV=\frac{\left(12691*K\right)}{\left(821.3+6.006*K\right)}$$

Equation (2) was used to determine the number of carbon double bond alkenes (*DB_average_*) per triglyceride molecule. In this equation, the *Nf* value is related to four methylene hydrogens of the glycerol backbone at 4.14 ppm and 4.29 ppm, divided by the number of related hydrogens [53].(2)$$DB_{average}=\frac{K}{2Nf}$$

After that, the IV and mol of alkenes in the castor oil were 102.2 g of I_2_ per 100 g of sample and 0.4026 mol per 100 g, respectively. The *DB_average_* per triglyceride molecule was equal to 3.3 bonds, which consisted of two ricinoleic acid chains and one linoleic chain.

### 2.3. Formulation and Photopolymerization

Six different formulations were evaluated in this work. The pristine formulation contains CO and 3SH in an equimolar (1:1 mol/mol) ratio considering the functional groups (C=C: SH); the BAPO was used as a photoinitiator (3 parts per hundred resin—phr), and this formulation was named CO3SH-P (P for pristine). After that, three formulations containing the hemp powder were prepared following the previous formulation, but adding 5 phr (CO3SH-H05), 10 phr (CO3SH-H10), and 20 phr (CO3SH-H20) of the filler. Last, two formulations containing the tagua and walnut shell powders were prepared using 10 phr of each (CO3SH-T10 and CO3SH-W10). All formulations can be seen in Table 1.

Subsequently, the formulations were added to a silicon support and cured under UV light (456 mW cm^−2^) using a Dymax ECE Flood lamp (Dymax Europe GmbH, Wiesbaden, Germany) for 120 s.

### 2.4. FTIR Analysis

Conversions were obtained by spreading the monomeric formulations in a SiC substrate at a thickness of 15 μm. Afterward, MIR FTIR spectra (4000–400 cm^−1^, resolution of 4 cm^−1^) were obtained at different curing times (0, 10, 20, 30, 60, 90, and 120 s) using a Nicolet iS10 spectrometer (Thermo Scientific, Milan, Italy) and analyzed using the OMNIC software V. 8.2.0.387. The curing occurred using a LightCuring LC8 (Hamamatsu Photonics, Milan, Italy), 365 nm, and an intensity of 80% (456 mW cm^−2^).

The conversions were determined using Equation (3) [54,55], monitoring the area of two bands (*A_group_*) at 1606 cm^−1^ and 2570 cm^−1^ related to C=C and S-H stretching, respectively. To normalize the spectrum, the band at 1741 cm^−1^ associated with C=O stretching was also monitored (*A_ref_*). The conversion was performed in triplicate.(3)C %=AgroupAreft=0−AgroupAreft=xAgroupAreft=0×100

The polymerization rate (*Rp*, mol s^−1^) was calculated using Equation (4) for each polymerization time (0, 10, 20, 30, 60, 90, and 120 s), providing the rate curve where [*n*_0_] is the mol concentration of thiol monomers, Δ_t_ is the difference between the two areas related to the S-H stretching, *A*_0_ is the initial area at a time equal to zero, and Δ_t_ is the time difference between the two analyses.(4)$$R_{p}=\left[n_{0}\right]\times \left[\frac{{∆}_{A}}{A_{0}*{∆}_{t}}\right]$$

### 2.5. Photo-Differential Scanning Calorimetry (Photo-DSC) and Dynamic Mechanical Analysis (DMA)

The photo-DSC curves for each formulation were obtained on a DSC-1 modulus (Mettler-Toledo, Milan, Italy). Approximately 8 mg of each sample was added to 40 μL aluminum crucibles, and the analyses were performed in isothermal mode (25 °C) under a nitrogen flow rate of 40 mL min^−1^. The samples were cured with a LightCuring LC8 (Hamamatsu Photonics), 365 nm, and an intensity of 80% (456 mW cm^−2^). The curve was obtained after two isothermal cycles to obtain only the total energy involved in the photopolymerization by subtracting the second curve (without any thermal event) from the first curve. After that, the polymerization rate (*R_p_*, mmol s^−1^) was calculated using Equation (5) [56], considering the time of exothermic polymerization peak (maximum value). The Δ*H_total_* is the enthalpy for a conversion of 100%, and *dH/dt* is the heat flow under a specific isothermal condition, in this case at 25 °C.(5)$$R_{p}=\frac{1}{{∆H}_{total}}x{\left(\frac{dH}{dt}\right)}_{T}$$

The DMA curves were obtained from Triton equipment. The UV-cured polymer samples had dimensions of 20 mm in length, 7 mm in width, and 0.5 mm in thickness, and were analyzed from −80 °C to 40 °C using a frequency of 1 Hz and a heating rate of 3 °C min^−1^. The DMA analysis provided the storage modulus (*E’*) and *tanδ* value curves. The glass transition temperature was defined as the temperature at the maximum loss factor (*tanδ*).

The crosslink density (*v*) was determined by Equation (6), where E’ is the storage modulus in the rubbery region, R is the gas constant (8.314 J K^−1^ mol^−1^), and T (*K*) is the rubbery temperature (*T_g_* + 50 °C).(6)$$v=\frac{E^{\prime }}{3RT}$$

### 2.6. Contact Angle, Bio-Based and Gel Contents, and Rheology

The contact angles for all UV-cured materials were measured using a Kruss DSA10 tensiometer (Krüss Scientific, Hamburg, Germany) equipped with a digital camera. A drop of water was deposited on the surface of each sample, and 10 optical scans were performed to verify the contact angle formed between the drop and the surface. This method was executed in triplicate.

Equation (7) allowed for calculation of the bio-based content (BC), in which the *m_CO_*, *m_3SH_*, and m_filler_ are the masses in grams of the castor oil, thiol monomers, and fillers, respectively. The *m_BAPO_* is the mass of the photoinitiator used that is not considered renewable.(7)$$BC\left(\%\right)=\left[\frac{m_{Co}+m_{3SH}+m_{filler}}{m_{Co}+m_{3SH}+m_{filler}+m_{BAPO}}\right]\times 100$$

For the gel content measurement, about 0.5 g (*m_i_*) of each sample was added to different metallic nets and submerged in chloroform for 24 h. All nets were dried at room temperature for 24 h, and the final mass (*m_f_*) was recorded in an analytical balance. The gel content was calculated using Equation (8).(8)$$GC\left(\%\right)=\left[\frac{m_{f}}{m_{i}}\right]\times 100$$

The viscosity measurements of the photocurable formulations were performed in an Anton-Paar—MCR302 Rheometer (Ganz, Austria) using a plate–plate configuration (metal-based disks). They were executed at room temperature using a shear rate from 0.01 to 1000 s^−1^.

### 2.7. Working Curves and 3D Printing

The working curves were executed for the pristine formulation (CO3SH-Pristine) and the formulations containing 10 phr of each filler (CO3SH-H10, CO3SH-T10, and CO3SH-W10) to obtain the curing depth (*C_d_*) and critical energy exposure, (*E_c_*) and consequently the minimum exposure time, for each layer in the 3D printing. These curves were performed in an Asiga Max X 3D printer (Asiga, Erfurt, Germany) using the maximum light intensity (*Emax* = 21.6 mW cm^−2^) at different times. The sample thickness was measured using a micrometer, resulting in the *C_d_* each time. The *Ec* was calculated using Equation (9), which is the critical energy for the formulation to reach the gelation state. The term *D_p_* is the depth of the resin when the light intensity is reduced to 1/e of its incident value (i.e., it can be obtained by the slope inclination of the analytical curve) [55,57].(9)$$C_{d}=D_{p}ln(\frac{E_{max}}{E_{c}})$$

The CO3SH-H10 was printed in the Asiga Max 3D printer (***E_max_*** = 21.6 mW cm^−2^, wavelength = 385 nm) at room temperature for two 3D structures (pyramid and honeycomb). The exposure time was 60 s for each layer (50 μm). After that, pieces were removed from the support and submerged in isopropyl alcohol for 5 min in an ultrasonic bath to remove the residual monomers. Then, the 3D structures were dried and post-cured for 30 min in Phrozen curing station equipment (wavelength = 405 nm).

### 2.8. Scanning Electronic Microscopy (SEM)

The morphology cross-sections for the UV-cured formulations, CO3SH-P, CO3SH-H10, CO3SH-T10, and CO3SH-W10, were observed in a JCM-6000 plus scanning electronic microscope (Jeol, Tokyo, Japan). The samples were placed in a cross-section holder, settled in a standard carbon adhesive, and covered with chromium. A low-pressure atmosphere was used, and the voltage was set at 5 kV.

## 3. Results and Discussion

### 3.1. Photocuring Process

The UV-curable formulation contains castor oil (CO) and trimethylolpropane tris(3-mercapto propionate) (3SH) as monomers and BAPO as a Type I photoinitiator; no solvent was used in this formulation. Normally, the 3SH is immiscible with vegetable oils—soybean, grapeseed, and so forth [34,35,36]. However, the mixture between the CO and 3SH is stable, since the CO presents polar domains in its structure (-OHs), which assist the interaction with the thiol group.

The thiol-ene photoinduced click reaction was exploited to overcome the crosslinking process. Finally, a crosslinked polymer containing sulfide moieties is formed [34,35,36]. The reactants and proposed reactions are displayed in Figure 1.

As mentioned in the introduction section, the reaction depends on the thiol and alkene concentrations, which should ideally be equimolar. Therefore, both groups were monitored by FTIR analysis to follow the curing process and obtain the polymerization conversion. The FTIR spectra at different times (Figure 2a) display a band at 2570 cm^−1^ related to S-H stretching (highlighted in orange) and a band at 1608 cm^−1^ associated with C=C stretching (highlighted in blue); the thiol-ene process can be confirmed as both bands diminish through irradiation time [54,55]. Moreover, the band at 3007 cm^−1^ is related to the C-H stretching of the alkene group, and it vanishes after the reaction [54,55]. It is possible to observe, from the conversion curves as a function of the irradiation time (Figure 2b), the same conversion trend for the thiol and alkyne groups. Both bands followed the same conversion tendency each time; therefore, the conversions at 10 s for S-H and C=C were 62 ± 4% and 56 ± 7%, respectively. At 120 s, the conversion by S-H was 89 ± 4%, and by C=C was 90 ± 6%. This clearly indicates the occurrence of click reactions with concomitant consumption of the thiols and double bonds (Figure 2c,d).

Thus, considering the same tendency, only the S-H band was followed to evaluate the conversion upon irradiation for the other formulations. A slight conversion decrease was observed when filler was added to the photocurable formulation (Figure 3a). For instance, the CO3SH-H05 (5 phr of filler) and CO3SH-H10 (10 phr) presented conversions of 84 ± 3% and 86 ± 3%, respectively. However, the conversion was practically the same between these two filled samples. Nevertheless, when a 20 phr filler amount was used, the conversion decreased to 74 ± 2% (CO3SH-H20). Therefore, the 10 phr was fixed as the maximum filler content. The conversion curves as a function of the irradiation time for the tagua (CO3SH-T10)- and walnut shell (CO3SH-W10)-filled formulations reached thiol conversions of 86 ± 4% and 87 ± 3%, respectively, after 120 s of irradiation.

Figure 3b shows the rates of polymerization for all samples. The fastest rate was observed for the CO3SH-P (without filler), reaching 1.11 mol s^−1^ at 10 s of polymerization. The second highest rate was observed for the CO3SH-H05, with 0.75 mol s^−1^, and the lowest for the CO3SH-T10 (0.47 mol s^−1^). Although the polymerization rates were different for each formulation, the conversions were similar.

These formulations also underwent a photo-DSC analysis (Figure 4a) to obtain the heat release, peak time, and polymerization rate at the peak. The pristine formulation (CO3SH-P) presented the highest heat release (168.4 J g^−1^) and the highest R_p_ (5.1 mmol s^−1^) at the peak (9 s). The filler incorporation into the polymeric matrix decreased the heat release and R_p_; therefore, the CO3SH-H20 presented the lowest values (44.2 J g^−1^ and 1.6 mmol s^−1^). However, comparing the formulations containing 10 phr, the CO3SH-H10 presented the highest heat release and R_p_ (55.9 J g^−1^ and 2.8 mmol s^−1^). The peak time did not change significantly in any sample. These findings confirm that the fillers affect the polymerization velocity negatively (Figure 4b), but the conversion is not affected at the end of the reaction, as confirmed by MIR analysis. Table 2 presents the conversion and rate of polymerization, heat release, and peak time for each formulation regarding the FTIR and DSC analyses.

### 3.2. Characterization of UV-Cured Materials

The bio-based content (BC) was calculated as a first step for all of the formulations investigated. Bio-based content is an important factor in determining the sustainability of final polymers. It considers the mass of renewable and non-renewable reactants in the final product and gives a percentage value. The higher the value, the higher its sustainability [9,58,59,60]. In this study, the CO, 3SH, and fillers are considered renewable, while the BAPO (photoinitiator) is considered a non-renewable compound. The amount of BAPO in the formulations was minimal (3 phr); therefore, all UV-Curable formulations presented a high BC value. The pristine formulation, CO3SH-P, presented a value of 97.0%, while the filled formulations with 10 phr of filler presented values of 97.3%.

The pristine formulation and the filled formulations up to a content of 10 phr were UV-cured and fully characterized. Gel content—GC—determines the quantity of residual uncrosslinked monomers in the final polymeric matrix (Figure 5a). The residual monomers are leached from the matrix by soaking in solvent. In an ideal matrix, the gel content should be 100%, indicating that all monomers were incorporated into the polymeric network [61]. The UV-cured pristine formulation, CO3SH-P, showed a higher gel content, reaching 99.2%, while the filled samples presented lower gel contents than the pristine one. Despite that, all samples presented values above 96.0%. The second highest was for the CO3SH-H10 (98.8%). These findings prove that the developed formulation for polymerization has another green and sustainable aspect, since the residual monomers and their release are minimal.

The contact angles for all UV-cured samples are displayed in Figure 5b. A contact angle higher than 90° indicates the sample is hydrophobic, since the water drop has less contact with the material surface [9,61]. In this sense, only the CO3SH-P presents this characteristic with a value of 95 ± 9°, and as expected, the filled polymeric matrixes presented values below 90°, since the filler is a bio-based cellulosic compound. The lowest contact angle was observed for the CO3SH-H10 (77 ± 9°), while the CO3SH-T10 and CO3SH-W10 presented values close to 90 °C. Therefore, the hemp bio-based cellulosic filler enhances the hydrophilicity of the polymeric network, which can be explained by a higher volume added of the hemp filler compared to the other fillers. In fact, although the same mass was added in each formulation (10 phr of filler), the hemp filler presented a higher volume; for comparison, all fillers with the same mass quantity (0.08055mg) can be seen in Appendix A.

The higher standard deviation in these composites can be attributed to the roughness, filler distribution, and orientation along the surface. When these factors are considered, the ideal surface is not reached; however, the trend effect of the filler can be considered [62,63,64].

A viscoelastic characterization of the UV-cured samples was performed using the DMA analysis. The tanδ and E’ curves are collected in Figure 6. It is possible to observe that the glass transition (T_g_) for the pristine UV-cured formulation was −17.6 ± 0.3 °C, while the UV-cured CO3SH-H10 showed a slight decrease with a value of −20.6 ± 1.1; for the crosslinked CO3SH-T10, a value of −15.8 ± 0.7 °C was registered, with a slight enhancement of the T_g_ value; and finally, the CO3SH-W10 showed a non-uniform tanδ peak, with a maximum cantered at −24.8 ± 0.4 °C and a shoulder to a higher temperature. Overall, there was no significant effect on the T_g_ of the crosslinked materials in the presence of the filler. Still, an enhancement of the modulus in the rubbery plateau was evident for the crosslinked network achieved in the presence of the hemp filler. The CO3SH-H10 UV-cured formulation showed a higher E’ value, which could be attributed to an important reinforcement effect due to the higher volume content of the filler in the photocurable formulation, which consequently also affects the crosslinking density by a better secondary interaction between the polymeric matrix and filler cellulosic structure. The crosslinked CO3SH-T10 and CO3SH-W10 both showed lower E’ values in the rubbery plateau compared with the crosslinked pristine CO3SH-P formulation. Table 3 presents the physical–chemical properties of all UV-cured formulations.

The SEM micrographs for the pristine formulation and filled polymers (10 phr) are present in Figure 7. These micrographs were taken from cross-sections of the fractured polymers to investigate the fracture points and internal morphologic features. All samples are similar, independent of the filler addition or source; this aspect is evinced by the fractural points (red arrow). However, the CO3SH-P shows more debris (blue arrow), indicating a more fragile structure. Lastly, the morphology of the surfaces exhibits roughness in all samples (green arrow).

### 3.3. Viscosity, Working Curves, and 3D Printing

Viscosity is a valuable parameter to be controlled for photocurable formulations to be exploited in 3D printing. Proper viscosity behavior guarantees correct distribution in the VAT and, consequently, correct layer formation, since the layering process depends on the monomers filling in the plate base. The ideal viscosity range is between 0.2 Pa·s and 10.0 Pa·s for printable formulations (green highlight in Figure 8a); therefore, the formulation should not present a lower or higher viscosity [55].

For this reason, rheological measurements were performed on the pristine formulation and the filled formulations containing 10 phr of the bio-based filler under shear rate. The CO3SH-P has a starting viscosity of 300 Pa·s; however, at a shear rate of 2 s^−1^, the formulation is within the perfect viscosity region. When the filler is added with a content of 10 phr, the viscosity reaches a value of around 1000 Pa·s at the starting point. However, at a shear rate of 7 s^−1^, all formulations are within the working region (dark green line). These findings prove that the formulations are suitable for 3D printing.

The working curves for the pristine and 10 phr formulations are displayed in Figure 8b. The first layer of the CO3SH-P was obtained after an irradiation time of 100 s to reach a thickness of 118 μm, differing from the CO3SH-H10, CO3SH-T10, and CO3SH-W10, which received thicknesses of 158 μm, 208 μm, and 121 μm in 50 s, respectively. The filler inside the formulation can enhance the polymerization process through light dispersion; this is verified by obtaining layers in only 25 s, and at this time, the CO3SH-T10 presents the highest thickness (64 μm). On the contrary, the CO3SH-W10 presented a layer thickness of 7 μm that can be justified by its dark color [52].

After 1500 s, the pristine formulation reached a thickness of 2180 μm, which was the highest one, followed by the CO3SH-T10 (1717 μm), CO3SH-H10 (1296 μm), and CO3SH-W10 (776 μm). The absence of fillers permits light to pass through the polymeric samples without dispersion and reach a higher thickness, since the pristine polymer does not show opacity. Still, as mentioned, the fillers are essential to get an ideal layer in a shorter time. The darker filler in the CO3SH-W10 also spoiled the maximum thickness of the final sample due to the higher opacity [52].

Although they had different thicknesses and times of gelation, all of the investigated photocurable formulations presented a critical energy (*E_c_*) of around 6.7 mJ cm^−2^, which demonstrated that the thiol-ene process for these formulations does not require higher energies to occur [55]. Table 4 contains all parameters from the working curves.

For the applicability test in 3D printing, the formulation CO3SH-H10 was selected due to its better viscoelastic performance, with a higher E’ value in the rubbery plateau and good values of *Cd* in the working curves with a few seconds of light irradiation. The first photo-printed structure was a pyramid (Figure 9a) with a decent shape and a yellow color. In the pyramid faces, three perfect circles could be seen as planned.

The honeycomb was selected for a more complex structure (Figure 9b), which presented a homogeneous shape and holes. This indicates that these formulations are feasible for use in 3D printing.

Some studies in the literature report the use of thiol-ene reactions in 3D printing, using non-renewable monomers such as liquid polybutadiene, siloxane monomers, thioester diallyl ether, Triallyl-1,3,5-triazine-2,4,6(1H,3H,5H)-trione (TTT), trimethylolpropane triacrylate, triethyleneglycol divinyl ether, and Tri(ethylene glycol) dimethacrylate (TEGDMA), producing yellow- to red-colored structures with good resolution and flexibility [65,66,67,68,69,70,71,72]. Therefore, the printable formulation in our study has structures and colors comparable to those of the non-renewables.

## 4. Conclusions

Greener solvent-free formulations from non-modified castor oil and trimethylolpropane tris(3-mercapto propionate) were prepared with different bio-based fillers and used for photopolymerization reactions. The polymerization occurred by a thiol-ene reaction in the presence of a photoinitiator under UV light, as confirmed by MIR analysis. The polymers without filler and those containing up to (10 phr) presented notable conversions close to 90% and fast reactions. Moreover, all polymers showed a bio-based content equal or superior to 97% and a gel content higher than 96%.

The filled formulation showed an excellent curing depth (*C_d_*) in a shorter time, indicating a high applicability for 3D printing. Furthermore, the critical energy, around 6.7 mJ cm^−2^, was calculated by the working curves, indicating that the reaction does not need high energy.

The formulation containing 10 phr of hemp filler was printed with perfect complex structures (pyramid and honeycomb shapes), proving the applicability of this study. These formulations can also be applied to flexible polymers, adhesives, and photocuring coatings. This study opens a new window for further studies using non-modified castor oil for covalent adaptative networks (CANs).

## Figures and Tables

**Figure 1 polymers-17-00587-f001:**
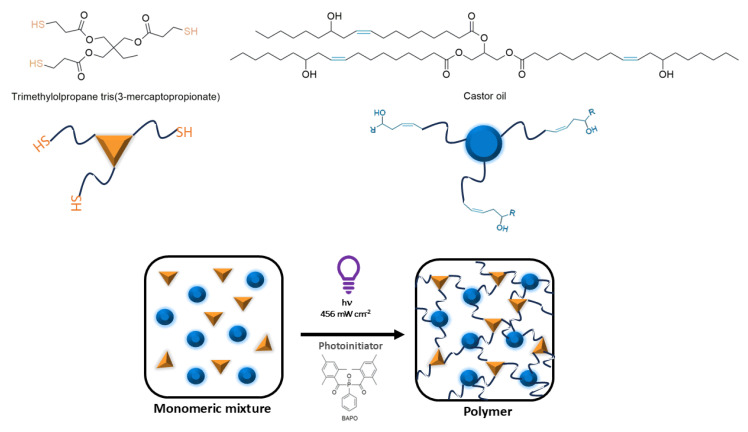
Chemical structure of monomers and photoinitiator used for thiol-ene formulation and photopolymerization reaction.

**Figure 2 polymers-17-00587-f002:**
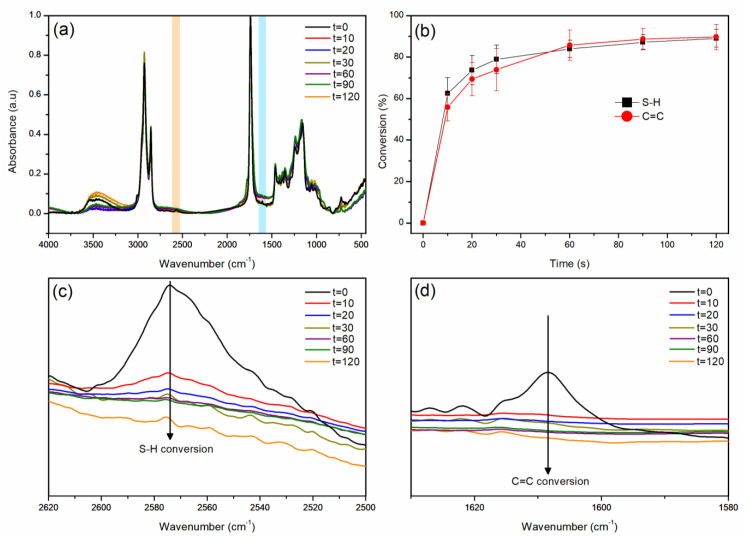
(**a**) FTIR spectra for CO3SH-P at different times highlighting the S-H band (orange area) and C=C band (blue area), (**b**) conversion by S-H and C=C functional groups, (**c**) amplified S-H band at different times, and (**d**) amplified C=C band at different times.

**Figure 3 polymers-17-00587-f003:**
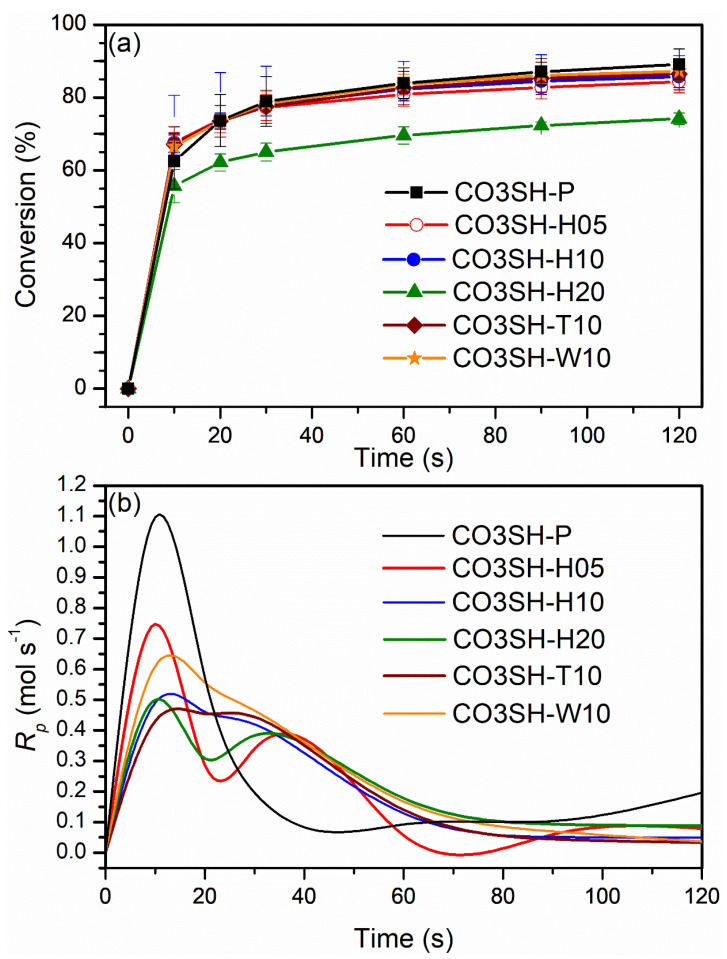
(**a**) Polymerization conversion for each formulation by MIR analysis and (**b**) respective polymerization rate.

**Figure 4 polymers-17-00587-f004:**
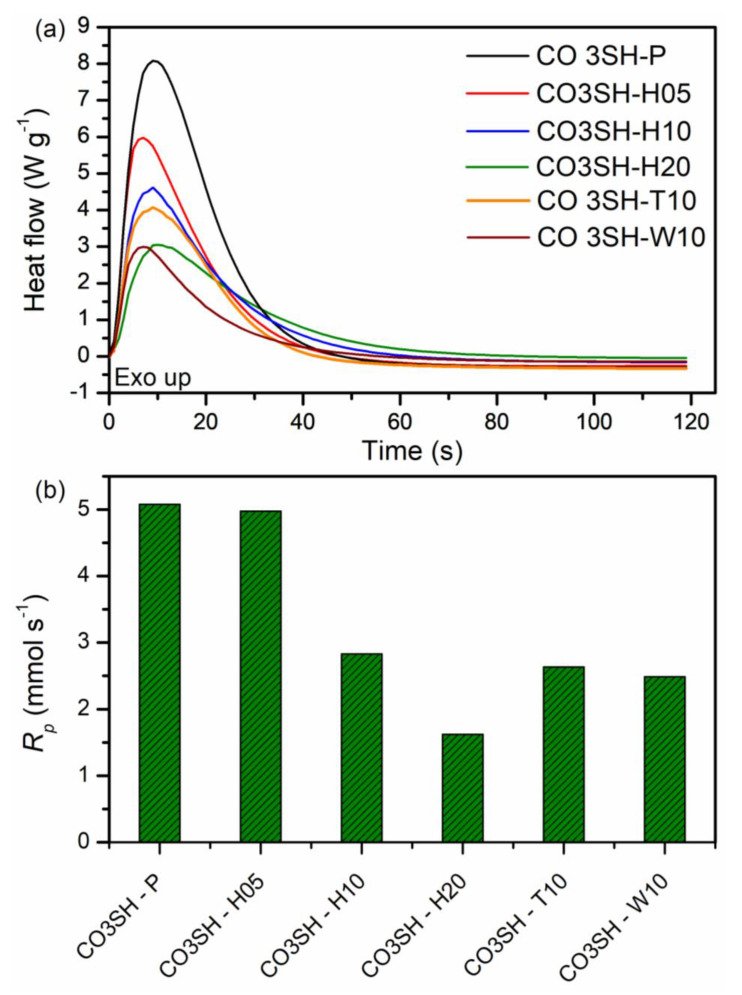
(**a**) DSC curves of polymerization for each formulation and (**b**) rate of polymerization calculated at peak maximum.

**Figure 5 polymers-17-00587-f005:**
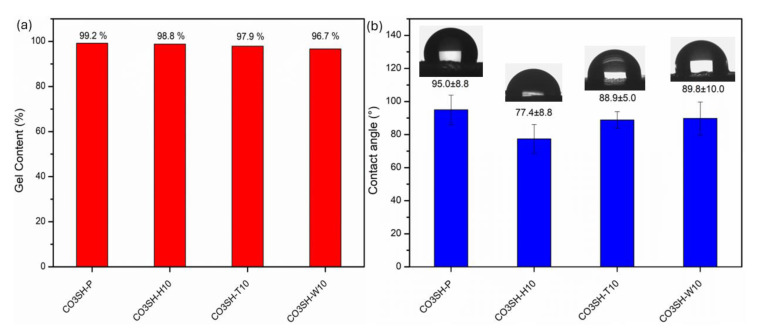
(**a**) Gel content and (**b**) contact angle for pristine formulation (CO3SH-P) and filled polymers containing 5 phr and 10 phr of bio-based cellulosic fillers.

**Figure 6 polymers-17-00587-f006:**
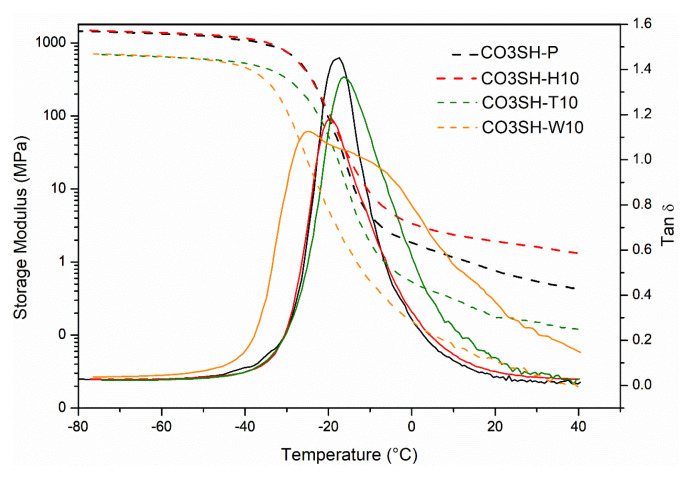
DMA relaxation analysis considering storage modulus (dashed line) and tanδ (solid line).

**Figure 7 polymers-17-00587-f007:**
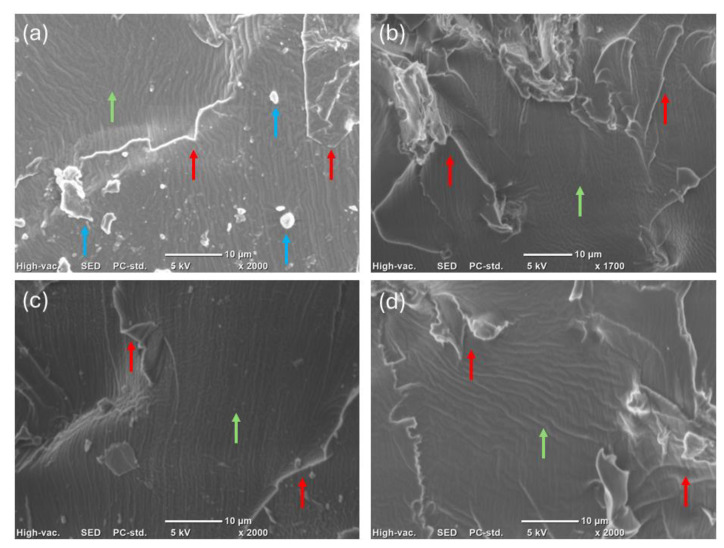
SEM micrographs for (**a**) CO3SH-P, (**b**) CO3SH-H10, (**c**) CO3SH-T10, and (**d**) CO3SH-W10.

**Figure 8 polymers-17-00587-f008:**
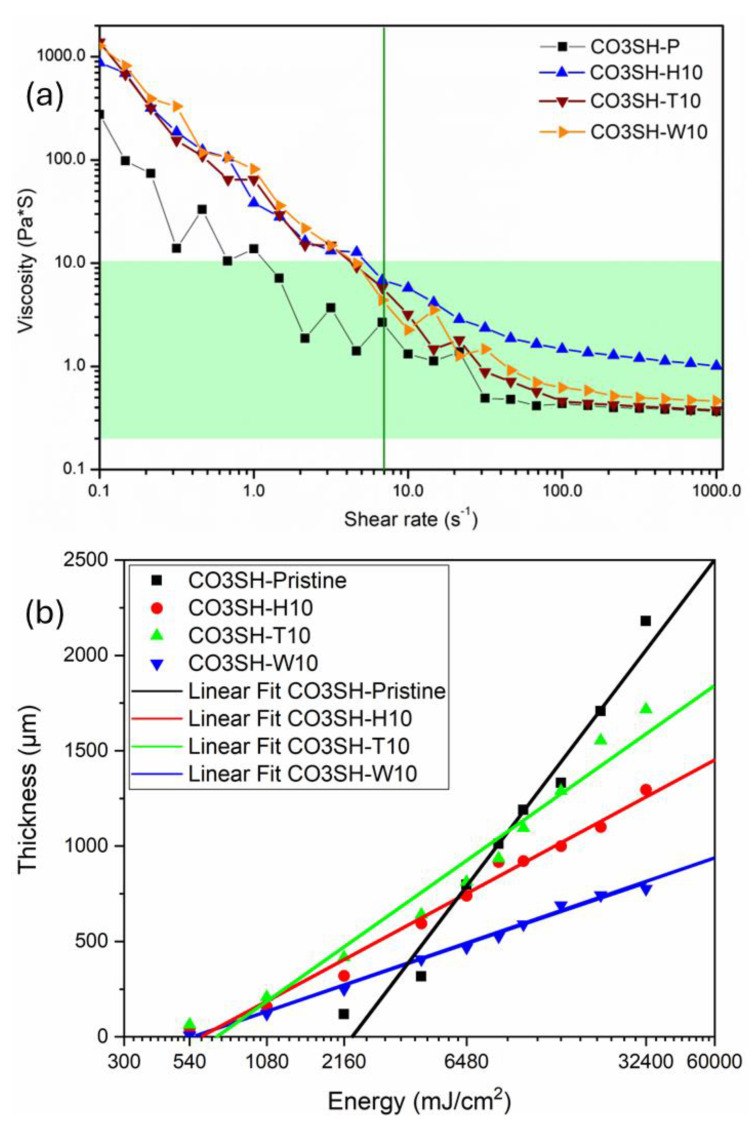
(**a**) Viscosity of each monomeric formulation in different shear rates and (**b**) working curves for CO3SH-P, CO3SH-H10, CO3SH-T10, and CO3SH-W10.

**Figure 9 polymers-17-00587-f009:**
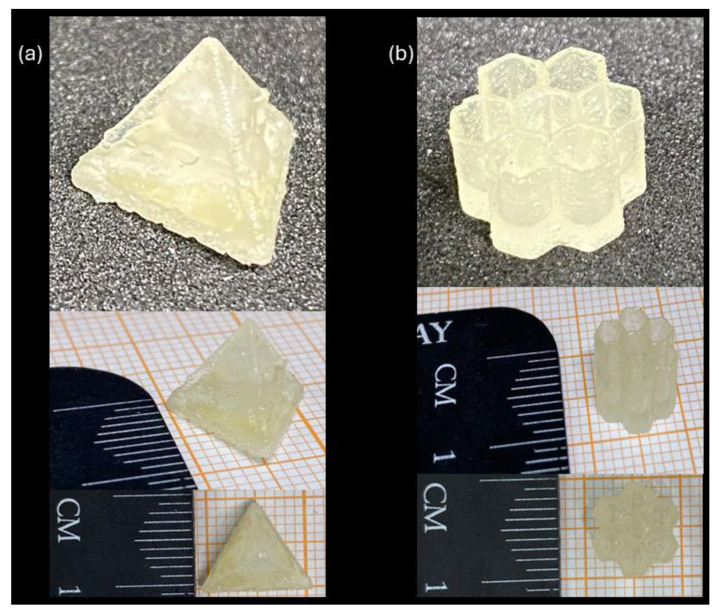
3D-printed samples using formulation CO3SH-H10: (**a**) pyramid and (**b**) honeycomb structure.

**Table 1 polymers-17-00587-t001:** Mass of castor oil (CO), Trimethylolpropane tris(3-mercapto propionate) (3SH), photoinitiator (BAPO), and filler for each formulation.

Formulation	CO (g)	3SH (g)	BAPO (g)	Filler (g)
CO3SH-P	0.5000	0.2815	0.024	-
CO3SH-H05	0.5000	0.2815	0.024	0.0403
CO3SH-H10	0.5000	0.2815	0.024	0.0806
CO3SH-H20	0.5000	0.2815	0.024	0.1612
CO3SH-T10	0.5000	0.2815	0.024	0.0806
CO3SH-W10	0.5000	0.2815	0.024	0.0806

**Table 2 polymers-17-00587-t002:** FTIR and DSC results regarding conversion (%), polymerization rate (*R_p_*), and enthalpy.

Sample	FTIR		DSC		
	C (%)	*R_p_* (mol s^−1^) at 10 s	Peak Time (s)	Δ*H* (J g^−1^)	*R_p_* (mmol s^−1^)
CO3SH-P	89 ± 4	1.1	9.0	168.4	5.1
CO3SH-H05	84 ± 3	0.7	7.0	126.5	4.9
CO3SH-H10	86 ± 3	0.6	9.0	101.2	2.8
CO3SH-H20	72 ± 1	0.7	10.0	80.1	1.6
CO3SH-T10	85 ± 4	0.5	9.0	93.7	2.6
CO3SH-W10	86 ± 2	0.8	7.0	67.2	2.5

**Table 3 polymers-17-00587-t003:** Physical–chemical properties of all UV-cured formulations.

	CO3SH-P	CO3SH-H10	CO3SH-T10	CO3SH-W10
BC (%)	97.0	97.3	97.3	97.3
GC (%)	99.2	98.8	97.9	96.7
Contact angle (°)	98.0 ± 8.8	77.4 ± 8.8	88.9 ± 5.0	89.8 ± 10.0
*T_g_*—DMA	−17.6 ± 0.3	−20.6 ± 1.1	−15.8 ± 0.7	−24.8 ± 0.4
Crosslinking density (mol m^−3^)	20.8 ± 1.5	155.7 ± 25.2	31.6 ± 1.0	7.2 ± 0.8

**Table 4 polymers-17-00587-t004:** Parameters were obtained by working curves for pristine formulation (CO3SH-P) and those filled with 10 phr (CO3SH-H10, CO3SH-T10, and CO3SH-W10).

Formulation	Time (s)	E (mJ cm^−2^)	C_d_ (mm)	ln E_c_ (mJ cm^−2^)
CO3SH-P	100	2160	118	7.73
	200	4320	317	
	300	6480	797	
	400	8640	1011	
	500	10,800	1190	
	700	15,120	1331	
	1000	21,600	1707	
	1500	32,400	2180	
CO3SH-H10	25	540	42	6.40
	50	1080	158	
	100	2160	320	
	200	4320	594	
	300	6480	740	
	400	8640	917	
	500	10,800	922	
	700	15,120	1000	
	1000	21,600	1101	
	1500	32,400	1296	
CO3SH-T10	25	540	64	6.53
	50	1080	208	
	100	2160	417	
	200	4320	641	
	300	6480	812	
	400	8640	936	
	500	10,800	1095	
	700	15,120	1289	
	1000	21,600	1554	
	1500	32,400	1717	
CO3SH-W1	25	540	7	6.34
	50	1080	121	
	100	2160	250	
	200	4320	407	
	300	6480	468	
	400	8640	527	
	500	10,800	590	
	700	15,120	690	
	1000	21,600	743	
	1500	32,400	776	

## Data Availability

The authors can confirm that all relevant data are included in the article.

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
