# Peer review of "Thiol-Ene Photopolymerization and 3D Printing of Non-Modified Castor Oil Containing Bio-Based Cellulosic Fillers"

_polymers, 2025, doi:10.3390/polym17050587_

Round 1

Reviewer 1 Report

Comments and Suggestions for Authors

This manuscript concerns the thiol-ene photopolymerization and 3D-printing of non-modified castor oil containing bio-based cellulosic fillers. Some corrections are needed before publication.

  1. The authors should discuss the effect of fillers that act as physical barriers and alter reaction kinetics or affect light penetration and radical formation.
  2. Authors should conduct an assessment on the long-term stability of printed structures.
  3. The results should be compared with similar articles.

Author Response

The answer to the reviewer comments have been uploaded in the file

Reviewer 2 Report

Comments and Suggestions for Authors

  1. The introduction is clear but the latest references are required to be incorporated
  2. The material and method section has too many sub-sections. Authors are encouraged to merge the sections such as sections 2.7, 2.8, and 2.9 can be merged to one section.  
  3. There is no need to add a border in the figures such as in fig. 7 and fig. 9
  4. Similarly figures 2c and 2d have colored backgrounds please remove the backgrounds.
  5. The abbreviation must be described in full form in the first instances
  6. Conditions for copolymerization must described in the method section such as temperature and light intensity 
  7. What is the rationale behind the use of these fillers please describe where appropriate. 

Author Response

The answer to the reviewer commensta have been uploaded in the file

Reviewer 3 Report

Comments and Suggestions for Authors

The work entitled “Thiol-ene photopolymerization and 3D-printing of non-modi-2 fied castor oil containing bio-based cellulosic fillers” submitted for publication in POLYMERS by Sangermano and cowors investigates bio-based photopolymer formulations incorporating cellulosic fillers (hemp, tagua, walnut) for thiol-ene UV-curable networks. It examines filler dispersion, photopolymerization kinetics, mechanical properties, and thermal stability. While bio-based content improves sustainability, challenges in filler compatibility, crosslink density, and mechanical performance require optimization for industrial applications.

Some sentences are overly complex or awkwardly phrased, affecting readability and clarity.

Some abbreviations are wrong, take for instance: Angew. Chemie - Int. Ed. And figure captions can make the manuscript appear less polished.

Small typo errors, such as "ocnversion" instead of "conversion,".

While bio-based cellulosic fillers are used, the discussion for selecting specific fillers (hemp, tagua, walnut) over other potential candidates is not deeply justified or stated. In addition, the study lacks a comparison with conventional petroleum-based photopolymers to benchmark the performance of bio-based formulations.

Overall, once the minor points are addressed the paper is ready for publication.

Author Response

(The authors gave the same response as above.)

Reviewer 4 Report

Comments and Suggestions for Authors

This manuscript presents a novel approach to developing sustainable, bio-based photopolymers using non-modified castor oil and cellulosic fillers for 3D printing. The work aligns with green chemistry principles and demonstrates promising results. However, several areas require clarification and expansion to enhance clarity, rigor, and impact. The comments for minor revisions are shown below:

  1. The Methods section is dense. Subheadings (e.g., 2.3 Formulation, 2.4 Characterization) would improve readability. Figures and tables are referenced but not visible in the provided text. Ensure all figures/tables are labeled correctly and legends are self-explanatory. Some of the figures are vague.
  2. Justify the choice of 3 phr BAPO photoinitiator. Is this concentration standard for thiol-ene systems? Provide references.
  3. Large standard deviations in contact angle measurements (e.g., 95±9°) suggest variability. Discuss potential causes (e.g., surface roughness, filler distribution).
  4. Compare conversion rates (70–90%) and mechanical properties with prior studies using modified castor oil derivatives. Highlight advantages of non-modified oil (e.g., reduced processing steps).
  5. Conclusions: Emphasize the novelty and suggest future directions. In addition, it is advised that the refined summaries should be better presented point by point.

Author Response

The answer to the reviewer have been uploaded in the file 

Round 2

Reviewer 1 Report

Comments and Suggestions for Authors

Accept in present form